# Stabilization of the Dimeric State of SARS-CoV-2 Main Protease by GC376 and Nirmatrelvir

**DOI:** 10.3390/ijms24076062

**Published:** 2023-03-23

**Authors:** Alessandro Paciaroni, Valeria Libera, Francesca Ripanti, Andrea Orecchini, Caterina Petrillo, Daniela Francisci, Elisabetta Schiaroli, Samuele Sabbatini, Anna Gidari, Elisa Bianconi, Antonio Macchiarulo, Rohanah Hussain, Lucia Silvestrini, Paolo Moretti, Norhan Belhaj, Matteo Vercelli, Yessica Roque, Paolo Mariani, Lucia Comez, Francesco Spinozzi

**Affiliations:** 1Department of Physics and Geology, University of Perugia, Via Alessandro Pascoli, 06123 Perugia, Italy; 2Istituto Officina dei Materiali-IOM, National Research Council-CNR, Via Alessandro Pascoli, 06123 Perugia, Italy; 3Department of Medicine and Surgery, Clinic of Infectious Diseases, University of Perugia, Piazzale Gambuli, 06129 Perugia, Italy; 4Department of Medicine and Surgery, Medical Microbiology Section, University of Perugia, Piazzale Gambuli, 06129 Perugia, Italy; 5Department of Pharmaceutical Sciences, University of Perugia, Via del Liceo, 06123 Perugia, Italy; 6Diamond Light Source Ltd., Harwell Science and Innovation Campus, Didcot OX11 0DE, UK; 7Department of Life and Environmental Sciences, Polytechnic University of Marche, Via Brecce Bianche, 12, 60131 Ancona, Italy

**Keywords:** main protease, dimerization, COVID-19, SARS-CoV-2, inhibitor, Paxlovid, PF-07321332, small angle X-ray scattering, microscale thermophoresis, circular dichroism

## Abstract

The main protease (Mpro or 3CLpro) is an enzyme that is evolutionarily conserved among different genera of coronaviruses. As it is essential for processing and maturing viral polyproteins, Mpro has been identified as a promising target for the development of broad-spectrum drugs against coronaviruses. Like SARS-CoV and MERS-CoV, the mature and active form of SARS-CoV-2 Mpro is a dimer composed of identical subunits, each with a single active site. Individual monomers, however, have very low or no catalytic activity. As such, inhibition of Mpro can be achieved by molecules that target the substrate binding pocket to block catalytic activity or target the dimerization process. In this study, we investigated GC376, a transition-state analog inhibitor of the main protease of feline infectious peritonitis coronavirus, and Nirmatrelvir (NMV), an oral, bioavailable SARS-CoV-2 Mpro inhibitor with pan-human coronavirus antiviral activity. Our results show that both GC376 and NMV are capable of strongly binding to SARS-CoV-2 Mpro and altering the monomer-dimer equilibrium by stabilizing the dimeric state. This behavior is proposed to be related to a structured hydrogen-bond network established at the Mpro active site, where hydrogen bonds between Ser1’ and Glu166/Phe140 are formed in addition to those achieved by the latter residues with GC376 or NMV.

## 1. Introduction

Coronaviruses, such as severe acute respiratory syndrome coronavirus 2 (SARS-CoV-2) and Middle East respiratory syndrome (MERS), can cause respiratory infections with a range of symptoms, from mild to severe. SARS-CoV-2, which is responsible for the COVID-19 pandemic, has led to millions of infections and deaths globally, as well as significant social and economic disruption [1,2]. Vaccines are a vital tool in the prevention of future infections but, even with a high immunity rate, a significant number of people will still need to receive therapeutic treatments. Antiviral research is a priority because it helps to reduce the severity of acute infections, and it is also necessary for people who are hesitant to get vaccinated or unable to receive vaccines for various reasons. The genome of SARS-CoV-2 encodes proteins including non-structural proteins (nsps), helicase, structural proteins, and accessory proteins. Mpro and the papain-like protease work together to process two polypeptides, polyprotein 1a and polyprotein 1ab, and release 16 nsps [3,4]. These nsps play important roles in replication, transcription, and virus recombination during infection. As inhibiting the protease catalytic function can halt the progression of COVID-19, and the Mpro three-dimensional structure is highly conserved among various coronaviruses [5], Mpro is a key target for the development of broad-spectrum antiviral drugs against COVID-19 [6,7] and other coronaviruses. Furthermore, Mpro inhibitors are unlikely to cause toxic effects because human proteases have different cleavage specificities.

Similar to SARS-CoV Mpro, SARS-CoV-2 Mpro forms dimers, with monomers having very low activity and dimerization being necessary for full enzymatic activity and virulence [8,9]. Each monomer is divided into three domains, as schematically shown in Figure 1. Domains I (residues 10–96) and II (residues 102–180) form a chymotrypsin-like folding scaffold containing six- and five-stranded antiparallel β-barrel structure, respectively [10]. The globular helical C-terminal domain III (residues 200–303), comprising a cluster of five α-helices connected to domain II by a long loop (residues 181–199), is key for the dimerization and the formation of a catalytically active Mpro. Within the binding site, four subsites can be identified: S1′, S1, S2, and S3/S4, which interact with the P1′, P1, P2, and P3 positions of the substrate, respectively [11]. The S1 subsite contains a catalytic dyad, which is composed of the Cys145 and His41 residues. The N-terminal 1–7 residues form the N-finger, which extends from domain I of one monomer to domain III of the other monomer and mediates multiple dimer interface interactions between domains I and II of both monomers. The interaction between the N-finger of each monomer and residues near the S1 substrate-binding subsite of the other monomer helps to stabilize the enzyme, correctly orient the S1 pocket of the substrate binding site, and increase catalytic efficiency [10].

The use of proteolytic inhibitors as antiviral therapeutics has been well-documented and has shown successful results [12,13]. Peptidomimetic inhibitors for the HIV protease [11] and small molecule inhibitors for the HCV protease [14] have been intensively tested. Proteolytic inhibitors, in combination with other drugs, are considered to play a crucial role in treating symptoms and reducing the spread of infection. Two crucial factors must be accounted for in the development of effective SARS-CoV-2 Mpro inhibitors. The first one is the direct interaction of the inhibitor with the catalytic site, which can be accomplished by utilizing molecules that target the substrate binding pocket. The second one is to reduce catalytic activity by using inhibitors that target the dimerization site. The latter approach is closely related to the equilibrium between the dimeric and monomeric state of Mpro in solution. Such thermodynamic equilibrium is quantified by the dimeric dissociation constant K_D_; however, a wide discrepancy among the different estimates of this parameter has been reported in the literature, with the values of K_D_ provided by various experimental techniques in a range from 230 ± 30 μM [15] down to 0.19 ± 0.03 μM [16]. Quite recently, by using small angle X-ray scattering (SAXS) we showed that K_D_ = 7 ± 1 μM for unbound Mpro, but this value was strongly dependent on the presence of small molecules interacting with the enzyme [17]. Understanding the structural details behind the interaction between Mpro and small molecules, and the way they are able to alter the monomer-dimer equilibrium, by disrupting or favoring dimerization, is key for the design and optimization of effective antivirals [18].

On these grounds, we decided to investigate the potential of GC376 and NMV, two inhibitors of the Mpro of feline infectious peritonitis coronavirus and SARS-CoV-2, respectively. GC376, which has been already shown to inhibit SARS-CoV-2 Mpro [19], is a bisulfite adduct prodrug of the corresponding aldehyde GC373 that forms a reversible covalent bond with the catalytic Cys145 and establishes H bonding interactions with the carboxyl group of Glu166, the carbonyl group of Phe140, and the imidazole of H163 [19]. NMV is a recently approved nitrile compound developed by Pfizer for human use as a covalent SARS-CoV-2 Mpro inhibitor [20,21]. The P1′ nitrile of NMV forms a reversible covalent thioimidate adduct with the catalytic Cys145, and H bonding interactions are established with Gln189, Phe140, Glu166, and His163 [22]. Despite detailed studies on the interaction of both GC376 and NMV with the Mpro binding site [22,23], little is known about their potential effect on the enzyme monomer-dimer equilibrium. By using SAXS we estimated, for the first time, the dimeric dissociation constant of Mpro in the presence of GC376 and NMV. By combining these results with other biophysical techniques, we found that GC376 and NMV not only strongly interact with SARS-CoV-2 Mpro but also have the capability to stabilize its dimeric form.

## 2. Results

### 2.1. MicroScale Thermoforesis

The microscale thermophoresis (MST) technique was used to determine the affinity between Mpro and six selected drugs. Thermophoresis is the directed movement of biomolecules in a temperature gradient, triggered by an infrared (IR) laser, which depends on molecular properties such as size, charge, hydration shell, and/or conformation that changes during a binding event [24]. In a typical MST experiment, 16 samples are simultaneously evaluated, where the concentration of fluorescent labelled protein is kept constant, and the concentration of unlabelled ligand varied through a large concentration range. Once the IR-laser induces a temperature gradient, the fluorophores in solution are excited and the migration of macromolecular complex induces a change in fluorescence signal. This variation is expressed as the normalized fluorescence (F_norm_) that is used to produce a binding curve and derive the dissociation constant (K_d_) of interaction. This method is characterized by a high sensibility and provides reliable and reproducible results [25]. To investigate protease-drug interaction, a constant concentration of protein is examined with a scalar concentration of GC376, carmofur, NMV, ritonavir, lopinavir, and nelfinavir. For three of six tested drugs, binding curves were produced, as shown in Figure 2, and K_d_ values derived. Specifically, GC376 and NMV bind Mpro at nanomolar concentrations (Table 1).

GC376 showed a K_d_ of 0.17 ± 0.04 µM for Mpro, in accordance with literature data where an IC50 of 0.19 ± 0.04 μM and a K_d_ value of 0.15 ± 0.03 μM are reported [23,26]. This compound was initially developed using structure-guided design to fight Middle East respiratory syndrome coronavirus (MERS-CoV) infections and, afterwards, it was identified as a covalent inhibitor of Nsp5, which stops the cleavage and the activation of functional viral proteins required for replication and transcription in host cells [27]. Carmofur is an antineoplastic able to bind the catalytic Cys145 and inhibit Mpro. In the MST experiment, it displayed a K_d_ value of 17 ± 4 μM toward the protease, in accordance with literature EC50 values in cells of 24 ± 3 μM [28]. NMV is an orally bioavailable Mpro inhibitor co-packaged with ritonavir, developed for the treatment and post-exposure prophylaxis of COVID-19 disease. It is a peptidomimetic that binds to the SARS-CoV-2 protease active site, blocking its activity with an inhibition constant (K_i_) value equal to 0.003 µM and a K_d_ value of 0.007 ± 0.004 μM [22,26,29]. In our experiment, this compound displayed a high affinity toward Mpro with a K_d_ of 0.004 ± 0.001 µM, in accordance with literature data. On the other hand, ritonavir, lopinavir, and nelfinavir do not bind Mpro effectively. Although the HIV protease inhibitors, specifically lopinavir and ritonavir, have shown in vitro activity against SARS-CoV MERS, they do not demonstrate a reliable Mpro inhibition activity and a clinical efficacy in the fight against SARS-CoV-2 [30].

### 2.2. Circular Dichroism and UV-Vis Absorption

Based on the MST results, we focused our attention on the GC376 and NMV compounds, which appear to interact very effectively with SARS-CoV-2 Mpro. Circular dichroism (CD) experiments were performed to further characterize the features of the Mpro-inhibitor complexes. Due to the chirality of GC376 and NMV, we compared the spectrum of the protein with that of the complex subtracted by the ligand signal. In the near-UV region, close to the broad peak centered at about 270–280 nm, the ellipticity increased when Mpro was in the presence of GC376, while the opposite happened when complexed with NMV (Figure 3A). These features indicate that both GC376 and NMV are able to bind Mpro and cause a slight rearrangement of the enzyme tertiary structure [31,32]. Indeed, the near-UV region provides information on the environment modifications around the aromatic amino acid side chains and on the dihedral angle of the disulphide bonds. Spectrum changes in this region were consistent with the scenario obtained by crystallographic data, in which the two ligands changed the environment around Phe140 [19,22].

As shown in Figure 3B, the binding interactions between Mpro and both GC376 and NMV were confirmed by UV-Vis melting experiments, proving that the enzyme undergoes unfolding at higher T_m_ in the presence of these two ligands. To obtain the thermodynamic parameters, we applied the two-state model already used in [17], where the dimer unfolds in two random-coil monomeric chains. The melting temperature of Mpro with 1% of DMSO was T_m_ = 54.5 ± 0.5 °C in agreement with [17]. The presence of GC376 increased the melting temperature by 2 degrees (T_m_ = 56.2 ± 0.4 °C), while NMV induces a stabilization of almost 10 degrees (T_m_ = 64 ± 1 °C). The NMV higher capability to stabilize the dimer structure is in accordance with its higher affinity for Mpro compared to GC376. We also note that the presence of the two drugs led to a less cooperative unfolding process.

### 2.3. Small Angle X-ray Scattering

Results of MST and CD indicate that GC376 and NMV strongly bind to Mpro. However, they give no clues on the way the ligands possibly alter the dimerization of the enzyme. Therefore, we exploited SAXS to determine whether GC376 and NMV are able to inhibit Mpro, not only by competing with the biological substrate but also by altering the dimeric enzyme functional form. SAXS has been already used to study the dimer-monomer equilibrium of the SARS-CoV and SARS-CoV-2 main proteases [17,33]. The advantage of this technique over other methods, such as analytical ultracentrifugation [10] or mass spectrometry [28], is that the samples are measured in solution and at equilibrium, so the conditions are closer to the physiological ones and kinetic effects can be neglected. From the scattering patterns shown in Figure 4, and applying the model described in the Materials and Methods section, we derived the dimeric dissociation constant of the Mpro alone K_D_ = 5 ± 1 μM, which is in agreement with that previously estimated [17].

As for the action of small molecules, according to the SAXS data shown in Figure 5, RTV may have a small effect on the balance between monomers and dimers by decreasing the K_D_ to 3 ± 1 μM. Based on simulations, it has been proposed that RTV may influence the formation and the stability of the Mpro dimeric state by interacting with a site near the domains III at the interface between the two monomers [34]. We found that GC376 and NMV have a much more significant impact on the monomer-dimer Mpro equilibrium, as they both strongly stabilize the dimeric form of the enzyme with K_D_ = 0.4 ± 1 μM and K_D_ = 0.7 ± 1 μM, respectively.

From the analysis of the SAXS patterns we derived additional thermodynamic parameters related to the dissociation process, also reported in Table 2. In more detail, the minor impact of the RTV inhibitor on K_D_ is a reflection of the relatively modest values of $∆H^{0}$ and $∆S^{0}$ compared to Mpro alone. In the case of GC376, the decrease of K_D_, i.e., the stabilization of the dimer, is mostly entropically driven. Indeed $∆S^{0}$ changed from −30 J/(K mol) in the absence of the inhibitor to −170 J/(K mol), whereas $∆H^{0}$ changed from a positive (20 kJ/mol) to a negative value (−15 kJ/mol), indicating major chemical stability of monomers with respect to dimers. This effect may be due to the formation of stronger hydrogen bonds within the monomer chain, or/and to the fact that the monomer can establish a more extended hydrogen bond network than the dimer with hydration water molecules. On the other hand, the overall entropic contribution to the dimer stabilization could be related to an increase of the disorder of the hydration water molecules released from the dimerization interface. The opposite happens for NMV, where we found that $∆H^{0}$ = 64 kJ/mol and $∆S^{0}$ = 100 J/(K mol) were both large and positive. In this case, the major stability of the dimer is mainly enthalpically driven, suggesting that stronger bonds are formed upon dimerization and that this contribution is not compensated by the increase of the disorder upon dimer dissociation. The dimer fractions as a function of total Mpro concentration for different temperatures are reported in Figure 6.

## 3. Discussion

From the MST method we derived values of the binding constant K_d_, which were in excellent agreement with those estimated by isothermal titration calorimetry for GC373, the reactive aldehyde form of GC376 (K_d_ = 0.15 ± 0.03 μM) [25], and NMV (K_d_ = 0.007 ± 0.004 μM) [26]. This confirmed that both the ligands firmly bind to the SARS-CoV-2 Mpro. SAXS results showed that both GC376 and NMV shifted the monomer-dimer Mpro equilibrium toward the dimeric state. Concerning GC376, this behavior matches a quite recent investigation where it was found to induce the dimerization of a predominantly monomeric mutant of the Mpro [26].

A possible explanation for the similar inhibitor-induced dimerization of Mpro by GC376 and NMV can be proposed on the basis of the structural properties of the complexes formed by the enzyme with the two ligands. A distinctive feature of Mpro dimer is the interaction of N-terminal residues (“N-finger”) of protomer A with residues of domain II of protomer B. In the dimer, the NH-group of Ser1 from protomer A forms strong H-bonds with the carboxylate group of Glu166 and the carbonyl of Phe140 of protomer B, and vice versa (Figure 7A). This interaction stabilizes the enzyme dimeric state and induces the cooperative formation of many hydrogen bonds, Gly138-His172, Ser139-Tyr126, Leu141-Tyr118, Gly143-Asn28, and Cys145-Asn28, which in turn promote the active conformation of the residues 138–145, called the C-loop as it contains the nucleophile Cys145 of the catalytic dyad [35]. Such an active conformation of the C-loop is key for substrate and inhibitor binding. In the presence of GC376, Glu166 and Phe140 of monomer B not only coordinate the N-finger of monomer A but also establish hydrogen bond with the P1 position of the drug, as shown in Figure 7B [19]. Quite interestingly, a similar scenario (see Figure 7C) can be proposed for Mpro in the presence of NMV starting from crystallographic data [36]. We suggest that the hydrogen bond network sustained by monomer A, GC376/NMV ligand, and monomer B in proximity of the N-finger terminal contributes to the increased stability of the Mpro dimeric state. The fact that both GC376 and NMV shift the Mpro monomer-dimer equilibrium toward the dimeric state may be related to the strong interaction of the two ligands with the Mpro active site.

Since the binding of both GC376 and NMV with the Mpro is also supported by the hydrogen bond network established between the C-terminal Ser1 of monomer A with Glu166 and Phe140 of monomer B, the bolstering of the Mpro dimerization could be a reflection of the very effective Mpro-ligand interaction. However, given the complexity of the process [17,26], establishing a causal relationship between the inhibition of the enzymatic activity and the dissociation of the dimeric form of Mpro by small molecules would require more experimental/numerical evidences. It is worth noting that the similar structural features of Mpro complexed with GC376 or NMV seem to be inconsistent with the different thermodynamic origin of dimer stabilization, which is entropically or enthalpically driven in the presence of GC376 and NMV respectively. However, it could be speculated that the specific ligand-dependent interaction of the Mpro monomer and dimer with their hydration water, already mentioned in the Results section, may explain this apparent inconsistency.

## 4. Materials and Methods

### 4.1. Sample Preparation

Mpro was expressed and purified as reported in [17]. The pGEX-6P-1 vector to encode the SARS-CoV-2 Mpro (NC_045512) was purchased from GenScript (clone ID_M16788F). For Mpro C-terminal His-tag removal, the Prescission (1 U for 100 µg of protein) cleavage reaction was performed at 4 °C for 4 h, and Prescission protease was then removed by a GSTrap FF column (GE-Healthcare, Chicago, IL, USA). The Mpro solution was further purified by FPLC size-exclusion chromatography on Superdex 75 10/300 GL column. Lopinavir, Ritonavir, Nelfinavir, Carmofur, GC376, and NMV inhibitors were bought from Selleck Chemicals GmbH, Planegg, Germany.

### 4.2. Microscale Thermophoresis (MST) Standard Assay

Recombinant coronavirus protease Mpro (Nsp5, 3CLpro) was fluorescently labelled to lysine residues with RED-NHS 2nd Generation dye supplied by NanoTemper Technologies (NanoTemper Technologies, GmbH, Munich, Germany) using a 1:3 protein/dye ratio. Briefly, 100 μL of 28 μM protein was mixed with 100 μL of 84 μM fluorophore and incubated for 30 min at room temperature in the dark, according to the instructions of the manufacturer. For the labelling process, a buffer composed of 130 mM NaHCO_3_, 50 mM NaCl, pH 8.2 was used. Unbounded dye was then removed by size-exclusion chromatography with the running buffer consisting of 20 mM Tris, 140 NaCl, pH 8.0 (Tris). Concentrations of labelled protein and RED-NHS dye were assessed using absorbance spectroscopy with a Thermo Scientific™ NanoDrop™ One spectrophotometer (Thermo Fisher Scientific Inc., Waltham, MA, USA). Specifically, these values were determined using a protein extinction coefficient $\varepsilon_{280nm}$ = 32,890 M^−^^1^cm^−^^1^, dye $\varepsilon_{650nm}$ = 195,000 M^−^^1^cm^−^^1^, a correcting factor at 280 nm equal to 0.04 and the following equations:(1)$$\left[protein\right]=\frac{A_{280}-(A_{650}\times 0.04)}{\varepsilon_{protein}\times l}$$
(2)$$\left[dye\right]=\frac{A_{650}}{\varepsilon_{dye}\times l}$$

The degree of labelling was determined as the ratio between the RED-NHS dye and protein concentration in the sample and was between 0.3–0.5.

To investigate the binding of nirmatrelvir (Cat. No. HY-138687, MedChemExpress LLC, Monmouth Junction NJ, USA) GC376 (Cat. No. S0475, Selleck Chemicals GmbH, Planegg, Germany), carmofur (Cat. No. C1494,Merck KGaA, Darmstadt, Germany ) ritonavir (Cat. No. S1185, Selleck Chemicals GmbH, Planegg, Germany), lopinavir (Cat. No. S1380, Selleck Chemicals GmbH, Planegg, Germany) and nelfinavir (Cat. No. S4282, Selleck Chemicals GmbH, Planegg, Germany) to Mpro, 16 pre-dilutions of compounds were prepared for MST experiments by 1:1 serial dilution in running buffer, such as Tris containing 4% DMSO, in PCR tubes to achieve a final volume of 10 μL. The molecules were tested starting from 1.25 µM for nirmatrelvir, 10 µM for GC376, 1000 µM for carmofur, 250 µM for ritonavir, 125 µM for lopinavir and nelfinavir. A 10 μL solution of RED-Mpro was added to each compound dilution and mixed to yield a final protein concentration of 20 nM for nirmatrelvir experiment and 50 nM for other molecules, 0.01% Tween20, 1 mM EDTA, 2% DMSO and an assay volume of 20 μL. After overnight incubation at +4 °C in the dark, the samples were loaded into premium-coated capillaries (MO-K025; NanoTemper Technologies, Munich, Germany) and inserted in the chip tray of Monolith NT.115 instrument (NanoTemper Technologies, Munich, Germany) for thermophoresis analysis, setting LED power at 40% and medium MST power. Recorded data were processed with NanoTemper MO Affinity Analysis v2.3 in manual mode, setting the hot region between 19/20 s. Confidence values (±) were indicated next to the K_d_ value for each of potential binders tested. Confidence values define the range where the K_d_ falls within a 68% of certainty, as declared by NanoTemper.

### 4.3. Circular Dichroism and UV-Vis Absorption

CD experiments were performed using JASCO J-810. The samples, Mpro, Mpro-GC376 (1:3 protein:drug molar ratio) and Mpro-NMV (1:3 protein:drug molar ratio), were measured in the near-UV range (250–340 nm) with a 0.5 cm cuvette cell pathlength and a protein concentration of 0.5 mg/mL and in 50 mM Tris buffer. A minimal amount of 1% DMSO was put in the buffer to solubilize the inhibitors. The melting studies were performed on the same samples with UV-Vis spectrophotometer JASCO V-500 across 26–74 °C temperature range with step of 2 °C.

### 4.4. Small Angle X-ray Scattering

SAXS experiments were carried out at the B21 beamline of the Diamond Synchrotron (Didcot, UK), operating with a fixed camera length (4.014 m) at 12.4 keV (λ = 1 Å) and with a flux of M∼10^12^ photons per second. Samples were injected in the capillary (thickness 1.7 mm) by means of a robotic apparatus and measured 21 times with an exposure time of 1 min. Data were normalized to the primary beam intensity, corrected for sample transmission, and calibrated to absolute scattering unit (cm^−^^1^) using water. A 20 mM Tris buffer, pH = 7.3, NaCl 100 mM, EDTA 1 mM, 1 mM DTT was used, with nominal monomer protein molar concentrations without inhibitors of 10, 20, and 30 µM and at the temperatures 15, 25, 30, 37, 40 and 45 °C. In the presence of inhibitors, SAXS curves were recorded at two Mpro monomer molar concentrations, 30 and 45 μM, and at the same temperatures as the Mpro alone.

### 4.5. Model for SAXS Data

The SAXS data analysis approach has been described in previous work [17]. Here we only summarize the main points. All SAXS curves were analyzed by a combination of the form factors of three particles: (i) the Mpro monomer, (ii) the Mpro dimer (*P*_1_(*q*) and *P*_2_(*q*), respectively) and (iii) of a tri-axial core-shell ellipsoid (*P*_el_(*q*)), with volume larger than 8 times the volume of the Mpro monomer, corresponding to 42,250 Å^3^, based on the idea that the protein may form multimers as large as an octamer [37]. Accordingly, the fitted macroscopic differential scattering cross section is:(3)$$\frac{d\Sigma }{d\Omega }\left(q\right)=N_{A}kC_{N}P\left(q\right)+B$$
where $N_{A}$ is Avogadro’s number, *κ* is the ratio between the total Mpro monomer molar concentration *C* = *κC_N_* (irrespective of their aggregation state) with respect to the nominal value *C_N_* = *c/M*_1_, *c* being the purified *w/v* protein concentration and *M*_1_ the molecular weight of Mpro monomer (*C* = *κC_N_*), and *B* is a flat background that accounts for possible uncertainties in the determination of transmissions of proteins and buffers samples. *P* (*q*) represents the average form factor of the system:(4)$$P\left(q\right)=\Phi \left[x_{1}P_{1}\left(q\right)+\frac{1}{2}\left(1-x_{1}\right)P_{2}\left(q\right)\right]+\frac{\left(1-\Phi \right)}{N_{agg}}P_{el}\left(q\right)$$
where Φ is the fraction of total Mpro monomers remaining in the monomeric state or forming dimers, *x_1_* is the fraction of Mpro the remains in the monomeric state in respect to monomer and dimers, and *N*_agg_ is the aggregation number of monomers forming the ellipsoids.

The form factors $P_{1}\left(q\right)$ and $P_{2}\left(q\right)$ were calculated from the crystal structure of Mpro dimer (6y2e.pdb) by using the SASMOL method considering one chain or both chains, respectively [17]. A unique value $d_{w}$ of the relative mass density of the shell water was optimized. The form factor of a tri-axial core-shell ellipsoid is:(5)$$P_{el}\left(q\right)=r_{e}^{2}\frac{2}{\pi }\int_{0}^{\frac{\pi }{2}}\int_{0}^{\frac{\pi }{2}}\sin\beta d\beta d\alpha {\left[\frac{4\pi }{3}\sum_{i=0}^{1}\left(A_{el}+k\delta \right)\left(B_{el}+k\delta \right)\left(C_{el}+k\delta \right)\left(\rho_{k+1}\;-\rho_{k}\right)\Phi_{S}(q{\left\{{[\left(A_{el}+k\delta \right)}^{2}\sin^{2}\alpha +{\left(B_{el}+k\delta \right)}^{2}\cos^{2}\alpha ]\sin^{2}\beta +{\left(C_{el}+k\delta \right)}^{2}\cos^{2}\beta \right\}}^{1/2})\right]}^{2}$$
where: $r_{e}=0.28\cdot 10^{-12}$ cm is the X-ray scattering length of the electron; $A_{el}$, $B_{el}$, and $C_{el}$ are the ellipsoid semi-axes (with the constraint $\left(4\pi /3\right)A_{el}B_{el}C_{el}=N_{agg}\;42250$ Å^3^), δ is the shell thickness fixed at 3 Å, and $\rho_{0},\rho_{1}$ and $\rho_{2}$ are the electron densities of the solvent, the shell and the core, respectively. The factor $\Phi_{S}\left(x\right)$ corresponds to the function describing the form factor of a homogeneous sphere, $\Phi_{S}\left(x\right)=(3/x^{3})(\sin x-x\cos x)$. According to known literature parameter, we fixed $\rho_{0}=\;0.33e/{Å}^{3}$, $\rho_{1}=0.33\;d_{w}e/{Å}^{3}$, $\rho_{2}=\;0.42e/{Å}^{3}$.

We assumed that monomer and dimer concentrations are regulated by the thermodynamic dissociation equilibrium constant:(6)$$K_{D}=\frac{{\left[M_{1}^{pro}\right]}^{2}}{\left[M_{2}^{pro}\right]}=\frac{2cx_{1}}{1-x_{1}}=e^{-\Delta G_{D}/RT}$$
where *x*_1_ is the molar fraction of proteins that remain in the monomeric state, Δ*G_D_* is the dissociation Gibbs free energy change, *R* is the universal gas constant, and *T* the absolute temperature. The fraction *x*_1_ can be calculated from Equation (6) at any Mpro concentration and temperature considering the dependency on temperature of Δ*G_D_* according to classical thermodynamics, $\Delta G=\Delta G^{0}+\left(\Delta C_{p}-\Delta S^{0}\right)\left(T-T_{0}\right)-\Delta C_{p}log(T/T_{0}),$ with $\Delta G^{0}=\Delta H^{0}-T_{0}\Delta S^{0}=-RT_{0}\log K_{D,0}$, where $\Delta G^{0},\Delta H^{0}$ and $\Delta S^{0}$ are the Gibbs free energy, the enthalpy and the entropy changes at the Mpro dissociation occurring at the reference temperature $T_{0}=298.15$ K, respectively, $\Delta C_{p}$ is the change of heat capacity at constant pressure, and $K_{D,0}$ is the dissociation constant at $T_{0}$. In this framework, all SAXS curves recorded at different temperatures, either without inhibitor or with each of three inhibitors, can be simultaneously fitted to the model by optimizing only three thermodynamic parameters. In our case we selected $K_{D,0},\Delta S^{0}$ and $\Delta C_{p},$ hence $\Delta H^{0}$ was a derived parameter.

The formation of aggregates of higher order than the dimer, seen as ellipsoids, is considered the result of an irreversible and uncontrolled process. Hence the fraction Φ, which regulates the ellipsoid concentration, as well as *N_agg_* and the semi-axes of the ellipsoids are treated as free parameters of each analyzed SAXS curve. All calculations were performed by the GENFIT software [38].

## 5. Conclusions

In this work, we characterized the ability of GC376 and NMV ligands to bind SARS-CoV-2 Mpro by using MST, providing that both establish strong interactions with the enzyme. CD spectroscopy revealed that these interactions reflect slight but appreciable modifications of Mpro tertiary structural features. SAXS experiments indicated that binding with GC376 and NMV inhibitors significantly strengthens the Mpro dimerization. We propose that this behavior involves the combined effect of the strong binding at the active site of GC376 and NMV together with the dimerization-related hydrogen bond network between the C-terminal Ser1 of monomer A with Glu166 and Phe140 of monomer B. On these grounds, the ability of the two potent inhibitors GC376 and NMV to promote Mpro dimerization may be related to their effectiveness at inhibiting the enzyme catalytic activity.

## Figures and Tables

**Figure 1 ijms-24-06062-f001:**
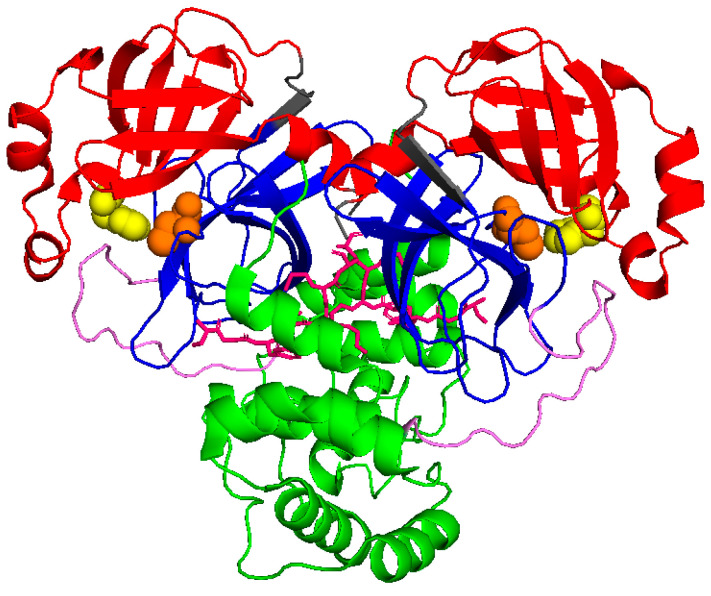
Schematic view of Mpro (6wtm.pdb): domain I is represented in red, domain II in blue, and domain III in green. N finger stick representation is colored in pink, while spheres identify the catalytic dyad formed by Cys145 and His41 (in orange and yellow, respectively).

**Figure 2 ijms-24-06062-f002:**
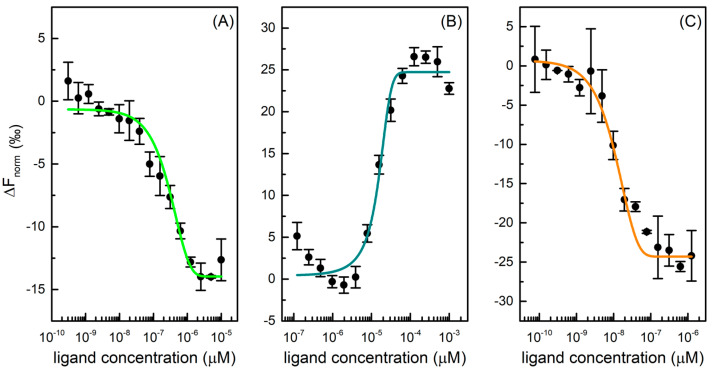
Dose-response curves for GC376 (**A**), carmofur (**B**), and NMV (**C**) binding to Mpro. The difference in normalized fluorescence [‰] is plotted against ligand concentration. Error bars represent the standard deviation of three independent measurements.

**Figure 3 ijms-24-06062-f003:**
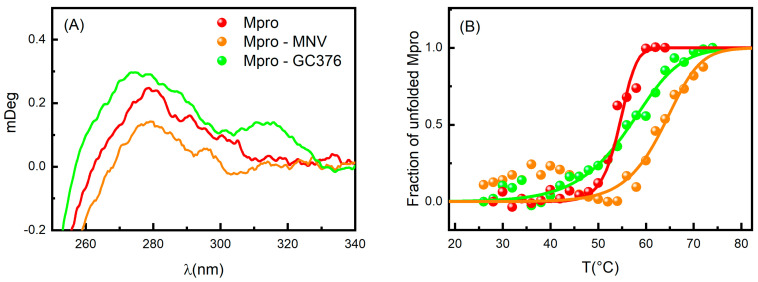
(**A**) CD spectra in the near-UV region of Mpro (red), Mpro-GC376 (green), and Mpro-NMV (orange). (**B**) Melting curves for the same samples obtained by UV-Vis absorption spectroscopy. The points are the normalized peak intensity at 280 nm.

**Figure 4 ijms-24-06062-f004:**
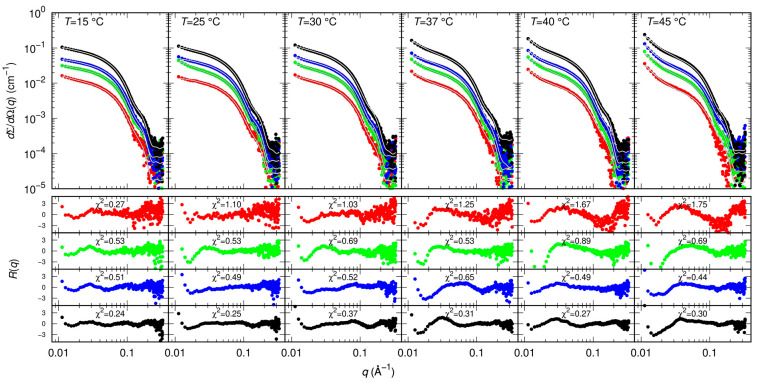
Measured SAXS intensity for different Mpro concentrations (10 μM red symbols, 20 μM green symbols, 30 μM blue symbols, 60 μM black symbols) and temperatures. Solid black/white curves represent fits to the data using model from Equations (3)–(5) of the Material and Methods section. Residual plots of each fit, defined as $R\left(q\right)={\sigma \left(q\right)}^{-1}[{d\Sigma /d\Omega }_{exp}(q)-{d\Sigma /d\Omega }_{fit}(q)]$, $\sigma \left(q\right)$ being the experimental uncertainty, are reported in the bottom panels.

**Figure 5 ijms-24-06062-f005:**
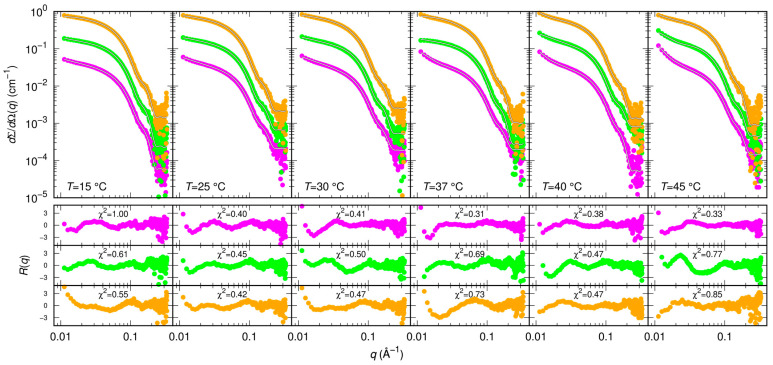
Measured SAXS intensity for Mpro in the presence of diverse inhibitors (RTV violet symbols, GC376 green symbols, NMV orange symbols) at 30 μM and at different temperatures. The molar ratio was 1:1. Subsequent curves are multiplied by a factor 4 for clarity. Solid black/white curves represent fits to the data using model from Equations (3)–(5) of the Material and Methods section. Residual plots of each fit, defined as $R\left(q\right)={\sigma \left(q\right)}^{-1}[{d\Sigma /d\Omega }_{exp}(q)-{d\Sigma /d\Omega }_{fit}(q)]$, $\sigma \left(q\right)$ being the experimental uncertainty, are reported in the bottom panels.

**Figure 6 ijms-24-06062-f006:**
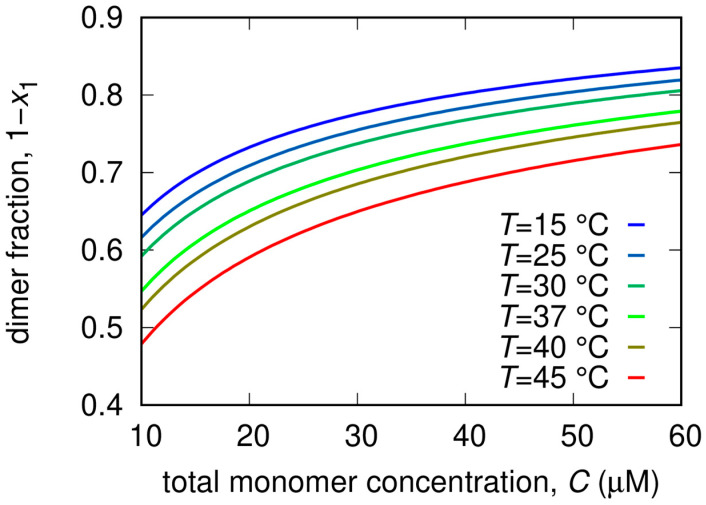
Fraction of dimers as a function of total Mpro concentration for different temperatures calculated on the basis of Equation (6) with the best fitting parameters (Table 2) derived by the analysis of SAXS data in the absence of inhibitors (Figure 3).

**Figure 7 ijms-24-06062-f007:**
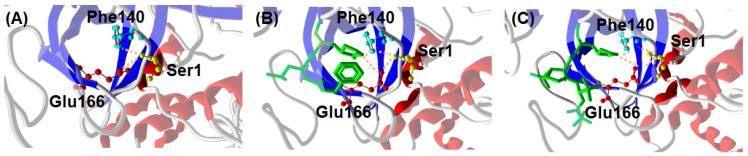
Interaction between C-terminal Ser1 of Mpro monomer A with Glu166 and Phe140 of monomer B from crystallography. (**A**) Mpro alone (6wtm.pdb), (**B**) Mpro with GC376 ligand (6wtj.pdb), and (**C**) Mpro with NMV ligand (7vh8.pdb).

**Table 1 ijms-24-06062-t001:** Dissociation constant K_d_ and confidence values of inhibitors with Mpro as estimated by MST measurements.

Compound	K_d_ (μM)
Lopinavir	>200
RTV	>250
Carmofur	17 ± 4
GC376	0.17 ± 0.04
Nelfinavir	>1000
NMV	0.004 ± 0.001

**Table 2 ijms-24-06062-t002:** Thermodynamic parameters derived from SAXS measurements.

Sample	$K_{D.0}$ (μM)	$\Delta H^{0}$ (kJ mol^−1^)	$\Delta S^{0}$ (J K^−1^mol^−1^)	$\Delta C_{p}$ (kJ K^−1^mol^−1^)
Mpro	5 ± 1	21 ± 4	−30 ± 10	1.3 ± 0.4
Mpro-RTV	3 ± 1	11 ± 5	−70 ± 50	4 ± 1
Mpro-GC376	0.4 ± 1	−15 ± 8	−170 ± 60	2.1 ± 0.8
Mpro-NMV	0.7 ± 1	64 ± 9	100 ± 30	−0.9 ± 0.8

## Data Availability

The data presented in this study are available on request from the corresponding authors.

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
