# Peer review of "Stabilization of the Dimeric State of SARS-CoV-2 Main Protease by GC376 and Nirmatrelvir"

_ijms, 2023, doi:10.3390/ijms24076062_

Round 1
Reviewer 1 Report
The SARS-COV-2 Main Protease MPro is responsible for processing two of the virus’ proproteins into several non-structural proteins. Inhibition of MPro therefore can halt the progression of COVID-19 which makes MPro an important target for the development of new antiviral drugs. Inhibition can be achieved by competitive inhibition of the active site or by disturbing the monomer-dimer equilibrium of the protease since only the dimer is active.
Paciaroni et al. biochemically investigate the effect of two known covalent inhibitors, GC376 and Nirmatrelvir, on the dimerization equilibrium of the SARS-COV-2 MPro. They report the inhibitors’ dissociation constants measured by Microscale Thermophoresis, near-UV CD measurements and show by UV melting and SAXS that both inhibitors stabilize the MPro dimer.
Even though MPro inhibition is of high interest, the presented data don’t seem very relevant for further drug or methods development.
1. It is not clear what the aim of this study is. From the introduction it sounds like the aim was disruption of the dimer interface because monomeric MPro is inactive (which is a valid strategy, but not for known covalent inhibitors, see point 3 below). However, the authors’ UV melting and SAXS data show that, in contrast to their initial hypothesis, both inhibitors stabilize the MPro dimer. While an explanation is presented based on the previously published crystal structures, it is not clear how MPro dimer stabilization in general benefits the development of new coronavirus drugs.
2. The crystal structures of MPro in complex with both inhibitors have been published (references 19 and 36, pdb 6wtj and 7vh8, also see Figure 5). Both complex structures show MPro as a dimer. Even though this does not allow conclusions about dimer stability, the structures do not indicate any features that would indicate a destabilization of the dimer. This does not justify the author’s initial hypothesis that the inhibitors in this study might disrupt the dimer and could be ‘dual inhibitors’, meaning competing with the substrate and inhibiting dimerization at the same time.
3. Both inhibitors are suicide inhibitors that covalently react with the catalytic Cys145. Therefore, binding is irreversible and any protein molecule that reacted with the inhibitor is unable to function, irrespective of the protein’s dimerization status. The KDs of the drugs lie in the nM range, while the dimerization of the protein occurs in the low µM range. Therefore, binding of the inhibitor and reaction with the catalytic Cys145 is already saturated before protein dimerization comes into play. Therefore, an additional disruption of the dimer not would bring any additional advantage (it could be relevant for reversibly binding, non-covalent inhibitors though, but this is not discussed either).
4. The SAXS method presented in this manuscript has been presented and used by the same group in a more extensive study (Silvestrini et al., reference 17). So there is no significant method development either.
5. Inhibitor KDs have been published previously, albeit using a different assay.
6. The presented CD data do not provide additional value considering that the complex structures are known and the CD does not explain anything about the monomer-dimer equilibrium.
Therefore, the presented manuscript lacks relevance, novelty and interest for a broad readership, and I consider it as not suitable for publication in IJMS.
Author Response
see the attached response

Reviewer 2 Report
The authors use an intriguing method to measure the intrinsic protein dimerization Kd using SAXS. SAXS, being a difference measurement, is limited in the range of sample concentrations which can be measured, due to low signal at low concentration, and aggregation (crowding) at higher concentration. Usually less than one order of magnitude in concentration can be measured. The authors overcome this limitation by adding temperature as a third dimension to the experiment and thereby using know thermodynamic relationships to explore a larger range of monomer/dimer ratios and in turn improving the accuracy of the measured Kd.
Figures 3 & 4. It is not easy to determine the goodness of fitting without either a Chi-squared value or better a Normalized Error plot (under the main plots). The Normalized Error plot can also determine if aggregation (Molecular Crowding) has altered the scattering formfactors as concentration or temperature are changed.
Alternatively, particularly for Figure 3, can you show a Michaelis-Menten plot of calculated Dimer Fraction vs. protein concentration.
Figure 5. Please use a white background for Molecular Graphics. Although most people use the PDF some will still want a printed copy. Sold black backgrounds are not friendly to printing and often leave ghost images on adjacent pages.
Please identify the Mpro catalytic residues either here in Figure 5 or in the text.
(257) “Our results indicate that neither GC376 nor NMV are effective as dual inhibitors, as their presence actually promotes dimerization.” A dual inhibitor would inhibit two things. The dual effect of inhibition and dimerization (which is normally activating) is not dual inhibition.
(329) It is never mentioned how/if the data are on a relative or absolute scale. Was a water or carbon film calibration used?
(331) (Didcot, UK)
Equation 4. How is the Pel(q) term determined? Surely this is an unknown and also dependent upon Nagg! It appears that you have 3 free parameters multiplied together (1-φ), Nagg, and Pel(q, Nagg). Given that (1-φ) is very small, this seems to be ill determined.
Note that a “tri-axial core-shell ellipsoid (Pel(q)), with volume larger than 8,343 times the volume of the Mpro monomer, corresponding to 42,250 Å3,” is a very ill-defined object.
I would accept that you are roughly modeling aggregation with an object of a certain size and shape, but this would seem to have too many parameters (rx, ry, rz, t-shell).
How are the P1, P2, and Pel form factors calculated? Are you using a program like Crysol or Foxes? If so which PDB entries are you using? These should be referenced.
What software is used to perform the multi-dimensional fitting? Mathematica, MATLAB, other?
Conclusions. You state that “we suggest that the effectiveness of covalent inhibitors like GC376 and NMV is tightly related to their ability to promote Mpro dimerization.” The inhibitors would seem to promote dimerization through their stabilization of the residues (140 & 166) and their interaction with Ser-1 of the adjacent molecule. However, the hydrogen bonds formed by Ser-1 can have a very small effect on dimerization, which involves a large surface area. To state a causal relationship between dimerization and inhibition would require more data. The stabilization of the monomer by these inhibitors will stabilize the protein, increasing its melting point and, unless sterically interfering with dimerization, promote dimerization by lowering the entropic contribution to the binding energy. This does not imply a direct relation between an inhibitors effectiveness and its ability to induce Mpro dimerization.
It is quite likely that their exist Mpro activators which promote Mpro dimerization without blocking the Mpro catalytic site. Dimerization of Mpro by a compound is not in itself an indicator of Mpro inhibition.
Author Response
see the attached response

Reviewer 3 Report
The article by Paciaroni et al. relates to the current debate concerning the use and the development of effective antiviral inhibitors in the treatment of COVID-19 infection. Proteins involved in biochemical pathways induced by viral infection (such as Mpro) have already been demonstrated to be interesting pharmaceutical targets. In fact, modulating or inhibiting these proteins can be effective in preventing disease evolution in patients who cannot be reached by vaccine, in severe cases of the disease, and to account for the genetic variability of SARS-CoV-2 virus. In line with the aim of highlighting structural and functional insights that act as modulators of the target protein-drug interactions, Paciaroni et al. use an integrated approach of biophysical techniques demonstrating that two antiviral molecules reported in literature exert their effect through selective stabilisation of the dimeric interface of Mpro protein. The outcome of this study may be of interest to the scientific community and may open up possibilities for investigation of new PPIs mechanisms for the inhibition of SARS-CoV-2 infection. It is my belief, however, that some changes are necessary to ensure that the message reaches the scientific community more effectively. In general, I believe that the arguments in favour of the presented mechanism are in need of some further elaboration. For example, CD analyses show a quite different structural effect upon the binding of the two drugs and the appearance of a band at 310 nm in the presence of GC376. Given that the two drugs activate a similar H-bond network, as claimed in the discussion, how do they explain this result? Moreover, the authors associate the conformational changes induced by drug binding with the increase in PPI at the dimer interface, so how do they explain that MNV is significantly more capable of stabilizing the dimer despite CD showing a smaller structural rearrangement at the binding pocket? Furthermore, since the binding pockets are exposed to the solvent, do the authors think that the dynamics of the water molecules could affect the proposed mechanism? Finally, I believe the reader would benefit from a general figure accompanying the structure description of the protein and drug binding sites (page 2, lines 66-70).
Author Response
see the attached response

Round 2
Reviewer 1 Report
Thank you for clarifying the scope of the present work and that the covalent binding of the inhibitors is reversible.
It still seems that the only novel statement is that dimerization of Mpro is strengthened, but I might indeed be missing some of the novelty and impact of the SAXS data. I do appreciate the added discussion of the thermodynamic parameters.
Therefore, if the other reviewers support publication, I will agree.